# Novel High-Pressure Nanocomposites for Cathode Materials in Sodium Batteries

**DOI:** 10.3390/nano13010164

**Published:** 2022-12-30

**Authors:** Aleksander Szpakiewicz-Szatan, Szymon Starzonek, Tomasz K. Pietrzak, Jerzy E. Garbarczyk, Sylwester J. Rzoska, Michał Boćkowski

**Affiliations:** 1Faculty of Physics, Warsaw University of Technology, 00-661 Warsaw, Poland; 2Institute of High Pressure Physics of the Polish Academy of Sciences, 01-142 Warsaw, Poland

**Keywords:** high pressures, nanocomposites, cathode materials, conductive glasses, hopping conductivity, Na-Ion batteries

## Abstract

A new nanocomposite material was prepared by high pressure processing of starting glass of nominal composition NaFePO${}_{4}$. Thermal, structural, electrical and dielectric properties of the prepared samples were studied by differential thermal analysis (DTA), X-ray diffraction (XRD) and broadband dielectric spectroscopy (BDS). It was demonstrated that high-pressure–high-temperature treatment (HPHT) led to an increase in the electrical conductivity of the initial glasses by two orders of magnitude. It was also shown that the observed effect was stronger than for the lithium analogue of this material studied by us earlier. The observed enhancement of conductivity was explained by Mott’s theory of electron hopping, which is more frequent in samples after pressure treatment. The final composite consisted of nanocrystalline NASICON (sodium (Na) Super Ionic CONductor) and alluaudite phases, which are electrochemically active in potential cathode materials for Na batteries. Average dimensions of crystallites estimated from XRD studies were between 40 and 90 nm, depending on the phase. Some new aspects of local dielectric relaxations in studied materials were also discussed. It was shown that a combination of high pressures and BDS method is a powerful method to study relaxation processes and molecular movements in solids. It was also pointed out that high-pressure cathode materials may exhibit higher volumetric capacities compared with commercially used cathodes with carbon additions.

## 1. Introduction

Development of novel functional materials plays a significant role in emerging better and better technologies that have a significant impact on our every-day lives. In particular, the green transformation of transport systems and power plants (from fossil fuels to zero emission), along with faster development of portable electronics, urges the development better green energy sources, including portable batteries, stationary energy magazines and fuel cells. Currently, the battery market is dominated by Li-ion cells, which are used in a wide variety of products, from small portable electronics to electric vehicles and trucks. On the one hand, the limited abundance of lithium is one of the limiting factors for further development of Li-ion technologies, and cathode manufacturing costs are the most important part of the total battery costs. On the other hand, large stationary energy storage facilities need to be built as essential parts of renewable power plants (solar or wind) whose energy production is very dependent on the weather conditions. Hence, interest in sodium superionic conductors has recently revived.

Mixed electronic–ionic (Na${}^{+}$) phosphate conductors are among the intensively studied candidates for Na-ion batteries. These include NASICON-type materials (e.g., Na${}_{3}$M${}_{2}$(PO${}_{4}$)${}_{3}$ [1] or Na${}_{3}$M${}_{2}$(PO${}_{4}$)${}_{2}$F${}_{3}$ [2,3], where M—transition metal ion, e.g., Fe, V, Mn, or Ti) and alluaudites (e.g., Na${}_{x}$M${}_{3}$(PO${}_{4}$)${}_{3}$ [4,5]). NaFePO${}_{4}$ is a sodium analogue of LiFePO${}_{4}$ olivine, proposed in 1997 by J.B. Goodenough [6]. It shares its lithium counterpart’s features, such as relatively high gravimetric capacity (154 mAh/g), abundant composition, low manufacturing costs and a low impact on the environment [7]. The olivine phase of NaFePO_4_ is unstable at ambient conditions. The stable phase of NaFePO_4_ is maricite (space group Pmnb), whose electrochemical performance is modest (e.g., [8]). However, it was discovered that NaFePO${}_{4}$ in an amorphous form can exhibit much better performance, i.e., 142 mAh/g [9,10].

Usually, phosphate cathode materials exhibit modest electrical conductivity, which may be a limiting factor for their performance in electrochemical cells. The most popular way of overcoming the limited conductivity is the addition of carbon during material syntheses or cathode preparation. The drawback of this approach is a decrease in the volumetric electrochemical capacity of such a cathode. Thermal nanocrystallization of glassy analogues of cathode materials may be an alternative way to significantly increase the electrical conductivity of cathode materials [11]. This approach consists of two steps: (i) synthesis of a glassy precursor; (ii) its proper heat-treatment to induce the appearance of nanocrystalline grains in a residual glassy matrix. The microstructure of such nanocrystalline materials provides favorable conditions for electronic hopping between aliovalent ions (e.g., Fe${}^{2+}$/Fe${}^{3+}$ or V${}^{4+}$/V${}^{5+}$). Application of this technique led to a giant increase in the conductivity in numerous phosphate glasses, including LiFe${}_{1-2.5x}$V${}_{x}$PO${}_{4}$ (i.e., LiFePO${}_{4}$ with partial substitution of Fe${}^{2+}$ with V${}^{5+}$), where nanocomposite materials with conductivity as high as ca 10 mS/cm at room temperature were obtained [12]. This phenomenon was qualitatively explained on the basis of Mott’s theory of polaron hopping [13,14]. It was also shown that the same approach led to a noticeable increase in the conductivity of sodium mixed conductors, i.e., NASICON-type [15] and alluaudite-type [16,17]. In case of LiFePO_4_, it was also experimentally proved that application of high pressure (of order of GPa) has a significant influence on electronic conductivity [18]. In compressed nanomaterials, the distances between the hopping centers are smaller than in pristine glasses, and hence irreversibly increase the electronic conductivity. In this work, we applied this approach to a sodium analogue of olivine, i.e., amorphous NaFePO${}_{4}$ doped or undoped with vanadium. The purpose of this work was to investigate the influences of high-pressure and high-temperature treatment (HPHT) on the electrical conductivity of this candidate for a cathode in sodium-ion batteries.

## 2. Experimental

### 2.1. Preparation of Glasses

Our initial aim was to prepare a glassy analogue of sodium olivine of nominal composition, NaFePO${}_{4}$. Being aware that preparation of pure glass of such composition may be difficult, we added some amount of V${}_{2}$O${}_{5}$ to selected samples as a supporting glass former. The nominal composition of such Na${}_{2}$O-FeO-V${}_{2}$O${}_{5}$-P${}_{2}$O${}_{5}$ heterogeneously doped glasses was NaFe${}_{0.85}$V${}_{0.10}$PO${}_{4}$.

To prepare the glasses, a melt quenching method, described by us elsewhere [18], was used. Appropriate amounts of: Na${}_{2}$CO${}_{3}$, FeC${}_{2}$O${}_{4}\cdot$2H${}_{2}$O, NH${}_{4}$H${}_{2}$PO${}_{4}$ and V${}_{2}$O${}_{5}$, were mixed in stoichiometric quantities and then finely ground in a planetary zirconia ball mill (Retsch GmbH, Düsseldorf, Germany). Next the substrates were put into a crucible, which was placed in a AFI-02 furnace (Argenta, Brzeziny, Poland). Samples were gradually heated up to 1523 K and then kept at this temperature to complete the calcination reaction related to release of volatile components. Finally, the molten glass was poured on the stainless steel or copper plate and immediately covered with a similar plate. It was noticed that quenching on copper was more effective because in such a case it was not necessary to dope NaFePO${}_{4}$ with vanadium to obtain a glass. An additional drawback of glasses doped with vanadium was their larger fragility compared to undoped glasses.

### 2.2. Experimental Methods

Differential thermal analysis (DTA), which allowed us to determine glass transition ($T_{g}$) and crystallization ($T_{c}$) temperatures of the studied glasses, was conducted using the TA Instruments Q600 SDT (Thermal Instruments, New Castle, DE, USA) with a heating rate of 10 K/min in the range of 323 to 1073 K in Ar flow.

X-ray diffraction (XRD) measurements was performed using Phillips X’PerT PRO, (Philips, Amsterdam, Netherlands) with Cu $K_{\alpha }$ (λ = 1.54 Å) in the range of Bragg angles from 5 to 115°(narrowed to the range 10–80° during analysis). Data were analyzed with the PANalytical High Score Plus software, ver. 4.7.0.24755, PANalytical B.V., Almelo, The Netherlands).

Electrical properties were measured with broadband dielectric spectroscopy (BDS) using a Novocontrol Alpha-A High Performance Frequency Analyzer (Novocontrol Technologies GmbH & Co. KG, Hundsangen, Germany) in the frequency range between 10 mHz and 1 GHz (narrowed to 100 mHz–10 MHz during analysis) in temperatures ranging from 123 to 473 K. The results were analyzed with the use of the Novocontrol WinFIT (WinFit Version 3.4 for 64 or 32 Bit MS-Windows 7, Vista, XP and 2000, Novocontrol Technologies GmbH & Co. KG, Hundsangen, Germany) with use of the multiple Havriliak–Negami functions.

The prepared NaFePO${}_{4}$ glasses were subjected to high-pressure–high-temperature treatment (HPHT), which was a key step in the present study. Developed in our laboratory, the HPHT method (Figure 1) is based on simultaneous action of isostatic high pressure (HP) and high temperature (HT). A sample of glass was put in the graphite crucible and then in a high-pressure chamber (Unipress Equipment, Warsaw, Poland) [19]. The chamber allowed us to regulate the pressure of inert gas (N${}_{2}$) and its temperature by use of a graphite resistor heater (Unipress Equipment, Warsaw, Poland). Temperature measurements were carried out by means of system of thermocouples, which allowed us to perform thermal analysis similarly to the DTA method (in this technique ΔT is defined as a difference between the temperature of the sample and the chamber).

In order to initiate nucleation of nanocrystallites from amorphous phase, the NF/co glass sample was subjected to 1 GPa pressure, then heated up to 974 K (i.e., above $T_{c}$ measured under pressure) and annealed at this temperature for 15 min. Next, the sample was cooled down to 873 K (i.e., between $T_{g}$ and $T_{c}$ measured under pressure) and annealed for a further 1.5 h in order to reduce entropy. Finally, it was cooled down to room temperature while being decompressed to atmospheric pressure. Our long-time monitoring showed that the effect of HPHT on samples properties (conductivity) lasted for months.

## 3. Results

### 3.1. Differential Thermal Analysis

Thermal analysis is a useful experimental technique because it gives information about the upper limit of thermal stability and phase transitions occurring in studied materials. Results of DTA are summarized in Figure 2 and Table 1.

Thermograms characteristic for glassy samples consist of endothermic shift corresponding to a glass transition and exothermic peak (or peaks) corresponding to crystallization of glass. For vanadium containing samples, a glass transition ($T_{g}$ = 683 K) and two crystallization peaks ($T_{c1}$ = 715 K, $T_{c2}$ = 767 K) were observed, regardless of the cooling technique (steel or copper) used during melt quenching. For NaFePO${}_{4}$ melt quenched between steel plates (st), no measurable glass transition nor crystallization peak was observed, which confirmed the crystalline state of that sample. On the other hand, a melt of the same composition quenched between copper plates (co) gave a glass with glass transition at about 776 K and only one crystallization peak at 844 K. It is worth noting at this point that: (i) a vanadium heterogeneous dopant helps to receive a glassy state in the case when steel plates are used (vanadium oxide plays a role of supporting glass former); (ii) a copper plate, due to its larger heat conductivity, is better, and vanadium doping is not necessary in that case; (iii) resignation from doping with vanadium and use of copper plates increases the thermal stability of the studied glasses from 683 to 776 K.

Curve e in Figure 2 shows DTA run corresponding to NF/co glass subjected to a high pressure. In this case, the values of $T_{g}$ and $T_{c}$ again increased, approaching the values 819 and 901 K, respectively, which widens the supercooled range.

### 3.2. X-ray Ray Diffraction

X-ray diffractogram of the sample after high-pressure–high-temperature treatment contains many peaks coming from basically two crystalline phases (curve e in Figure 3). By comparing patterns (c) and (e) in Figure 3, one can notice the appearance of new crystalline phases after HPHT treatment not observed in pattern c.

The singular peaks visible at curve a in Figure 3 may have originated from vanadium iron oxide and iron oxide crystalline nuclei. Curve b in Figure 3 shows improvement of quality in glassy material obtained from NaV${}_{0.10}$Fe${}_{0.85}$PO${}_{4}$ with application of the copper plate. This confirmation led us to synthesis of pure glassy NaFePO${}_{4}$ (curve d in Figure 3) which was used in further experiments.

The diffractogram of NF/st sample (curve c in Figure 3) shows peaks associated mainly with the NaFePO${}_{4}$ of maricite structure, which exhibits the same composition as olivine, but its electrochemical properties are modest. In contrast to olivine (orthorhombic system), maricite exhibits a tetragonal structure. The difference between maricite and phospho-olivine is swapped locations of alkali metal (sodium or lithium) and iron. While olivine exhibits tunnels of alkali metal ions, maricite lacks those channels. This explains the difference in ionic conductivity of those two polymorphs and the difference in their ability for an Na${}^{+}$ intercalation/deintercalation reaction.

A diffractogram of the sample after HPHT treatment (curve e in Figure 3) indicates a presence of electrochemically active phases such as Na${}_{3}$Fe${}_{2}$(PO${}_{4}$)${}_{3}$ (NASICON structure) and Na${}_{2}$Fe${}_{3}$(PO${}_{4}$)${}_{3}$ (alluaudite structure).

Alluaudite’s structure is monoclinic (space group I2/b). In this structure, easy conduction pathways for sodium ions can be observed, which explains the better electrochemical ability of alluaudite compared with maricite. The second identified phase—NASICON—exhibits a rhombohedral structure (space group R-3c). In this structure, similarly to alluaudite, sodium ion channels facilitating ion mobility exist.

The difference between sodium-to-iron ratios in alluaudite (2:3) and NASICON (3:2) phases could explain their coexistence in an approximately 1:1 ratio (more precisely 9:11) in the final composites (in substrates of reaction, the ratio of sodium to iron ratio was 1:1).

Using software associated with the used diffractometer, it was possible to estimate (based on Scherrer formula) the average sizes of grains detected in the identified crystalline phases (Table 2). In the NF/st sample, only maricite phase crystallized, whereas NF/co was a purely amorphous, glassy material. Nanomaterials obtained after HPHT treatment were composed of crystallites with average sizes of 91 nm (NASICON) and 42 nm (alluaudite).

### 3.3. Broadband Dielectric Spectroscopy

#### 3.3.1. Temperature Dependencies of Electric Conductivity

Broadband dielectric spectroscopy gives wide opportunities for dielectric and electric studies of various systems. A starting point is complex electrical permittivity $\varepsilon^{*}=\varepsilon^{\prime }\left(\omega \right)+i\varepsilon^{\prime \prime }\left(\omega \right)$, whose imaginary part is responsible for electrical conduction by the formula: (1)$$\sigma^{\prime }=\sigma_{DC}+\omega \varepsilon_{0}\varepsilon^{\prime \prime }\left(\omega \right)$$

In this section, we concentrate on dc conductivity determined from the real part of each complex conductivity spectrum. Figure 4 shows temperature dependencies of conductivity for the studied samples before and after high-pressure–high-temperature treatment (HPHT). Results could be compared with our earlier findings concerning LiFe${}_{0.75}$V${}_{0.10}$PO${}_{4}$ olivine-like cathode materials [18]. As was already mentioned, cathode materials should exhibit mixed electronic-ionic conductivity with distinct predominance of the electronic component. In fact, conductivity runs presented in Figure 4 relate to total electric conductivity, but the ionic component is practically fully masked by the strongly dominant and much more frequent electronic hopping between Fe${}^{2+}$–Fe${}^{3+}$ redox couples via oxygen bridge [11].

All experimental dependencies shown in Figure 4 can be described by well-known Mott’s formulas referring to electron transport in glasses and disordered systems [13,14]. Formulas given below are valid for two different temperature ranges, characterized by Debye temperature $\theta_{D}$, known from the theory of heat capacity.

For $T>\frac{\theta_{D}}{2}$, electron hops are phonon assisted (the jumps are facilitated by vibration modes), and the temperature dependence of electronic conductivity is given by the following Arrhenius-like expression.
(2)$$\sigma \left(T\right)T=\nu_{el}c\left(1-c\right)\frac{e^{2}}{Rk_{B}}\exp\left(-2\alpha R\right)\exp\left(-\frac{E_{a}}{k_{B}T}\right)$$
where *R* is the average distance between hopping centers (Fe${}^{2+}$/Fe${}^{3+}$), $k_{B}$ is the Boltzmann constant, *T* is temperature, $\nu_{el}=\frac{h}{m_{e}R^{2}}$, α is the inverse of localization length of the electron wave function, *c* is the fraction of occupied hopping sites for electrons and $E_{a}$ is the activation energy [13,14]. This kind of electronic transport is usually referred to as hopping of small polarons, because jumps of localized electron in an ionic structure is coupled with simultaneous motion of elastic deformation of local (small) surroundings [11].

For temperatures as low as $T<\frac{\theta_{D}}{4}$, when phonons are “frozen”, the electron transport mechanism changes to the variable range hopping, which follows the famous “$T^{\frac{-1}{4}}$” formula. In this temperature regime, jumping electrons are looking for states of similar energies, which do not necessarily correspond to adjacent sites. The experimental dependencies of conductivity (for glass and glass-ceramics) deviate at low temperatures from the Arrhenius behavior and follow the formula: (3)$$\sigma \left(T\right)=A\exp\left(-BT^{\frac{-1}{4}}\right)$$
where *A* and *B* are defined in [13,14]. Two aforementioned temperature ranges of Mott’s theory are visible in Figure 4. This kind of behavior we also observed in our earlier papers [11,20]. One can see from Figure 4 that HPHT treatment considerably enhances electric conductivity (about two orders of magnitude) from $3.18\cdot 10^{-10}$ to $3.58\cdot 10^{-8}$ S/cm at room temperature (and from $2.05\cdot 10^{-8}$ to $1.17\cdot 10^{-7}$ S/cm at 100 ${}^{\circ }$C) and decreases the value of activation energy by 28%. Additionally, it is worthy noting that the increase in conductivity for Na composites was higher than for their Li analogues [18]. For comparison, room temperature conductivities of NASICON and tri-iron alluaudite glass were equal to $1.33\cdot 10^{-11}$ S/cm [15] and $2.87\cdot 10^{-9}$ S/cm [21], respectively.

#### 3.3.2. Temperature Dependencies of Relaxation Times

Electric conduction is a macroscopic phenomenon, and measurement of dc conductivity may not provide microscopic insight into the processes occurring. Separation of component processes contributing to the total conductivity and dielectric response is possible thanks to various spectroscopic representations. In this study, we used analysis of electric modulus interrelated with complex electrical permittivity by the following: (4)$$M^{*}=\frac{1}{\varepsilon^{*}\left(\omega \right)}=M^{\prime }\left(\omega \right)+iM^{\prime \prime }\left(\omega \right)=\frac{\varepsilon^{\prime }\left(\omega \right)}{\varepsilon^{\prime 2}\left(\omega \right)+\varepsilon^{\prime \prime 2}\left(\omega \right)}+i\frac{\varepsilon^{\prime \prime }\left(\omega \right)}{\varepsilon^{\prime 2}\left(\omega \right)+\varepsilon^{\prime \prime 2}\left(\omega \right)}$$

The imaginary part of electric modulus $M^{\prime \prime }\left(\omega \right)=\frac{\varepsilon^{\prime \prime }\left(\omega \right)}{\varepsilon^{\prime 2}\left(\omega \right)+\varepsilon^{\prime \prime 2}\left(\omega \right)}$ gives information on the relaxation processes related to the transport of various charge carriers in the studied materials.

Figure 5 and Figure 6 present deconvoluted spectra of imaginary parts of the electric modulus. The straight parts correspond to dc conductivity, whereas the peaks apparently relate to relaxation times: of ion hopping ($\tau_{ih}$), electron hopping ($\tau_{eh}$) and ion motion ($\tau_{\sigma }$) through grain boundaries induced by HPHT. An almost invisible effect, resembling intercluster transport, corresponding probably to small heterogeneity, was also observed in starting glasses. It was taken into consideration to improve the fit of the spectra before HPHT. Different mechanisms of ion transfer in solids are broadly discussed in [22,23,24,25]. Transport of ions and electrons related with dc conductivity describes long-range properties of the material. On the other hand, hopping (which is also contributes to dc conduction) is mainly related to ac conductivity and might be described by temperature-activated processes.

Taking into consideration common properties of cathode materials, one can postulate that low-frequency processes resolved from spectra in Figure 5 correspond to ionic conduction, and high frequency ones refer to electronic conduction. In fact, ionic conductivity in the studied materials is very low (jumps of Na${}^{+}$ are slow), and electronic conduction predominates, because hopping of electrons is much more frequent. Further conclusions resulting from Figure 5 are as follows: (i) HTHP treatment leads to a considerable shift in modulus peaks towards much higher frequencies; (ii) the appearance of a low-frequency arm is correlated with pressure-induced crystallization of the studied samples.

Resolved spectra allowed us to determine relaxation times corresponding to related processes. According to the widely accepted Havriliak–Negami relation [26,27], the following relationship was assumed between frequency ($f_{max}$) maximum and relaxation time τ: (5)$$\omega_{max}=\tau^{-1}{\left(\frac{\sin(\frac{\alpha \pi }{2\left(\beta +1\right)})}{\sin(\frac{\alpha \beta \pi }{2\left(\beta +1\right)})}\right)}^{1/\alpha }$$
where α and β are parameters of the H-N relaxation element. By taking into account a frequency range (and values of H-N parameters) in our measurements, it was possible to approximate this formula into a simpler form: $\tau ={\left(2\pi f_{max}\right)}^{-1}$.

The calculated values of relaxation time τ plotted against an Arrhenius temperature scale are given in Figure 7. Pressure-induced reduction in relaxation time was observed in the whole temperature range. Before HPHT treatment, electron and ionic hopping were thermally activated, exhibiting a single activation energy. After HPHT, temperature dependencies of relaxation times became non-Arrhenius in some ranges, apparently due to heterogeneity caused by the crystallization of two new phases. This is clearly visible for relaxation process related to inter-grain transport of ions (green curve in Figure 7).

By comparing Figure 4 and Figure 7 and taking into account only electron hopping, it can be seen that the corresponding values of activation energies determined from temperature dependencies of relaxation time (0.54 eV, 0.39 eV) and electric conductivity (0.54 eV, 0.40 eV) are the same within limits of experimental uncertainty. It is also clear that HPHT treatment disturbs Arrhenius behavior for temperatures higher than 350 K and changes the mechanism of transport for temperatures lower than 220 K (Figure 7).

## 4. Discussion

Hopping of electrons required adjacent Fe${}^{2+}$/Fe${}^{3+}$ centers. On the other hand, hopping of ions requires empty sites in the vicinity of given Na${}^{+}$ cations. After HPHT treatment, a ramified network of grain boundaries appeared. Defective surfaces of those boundaries are a source of aliovalent iron centers. Therefore, in such conditions, transport of electrons occurs mainly along grain boundaries (highly disordered nanocrystallite shells), and transport of ions occurs across grain boundaries (inter-grain) and in the bulk of crystallites (intra-grain). Taking into account experimental data obtained in this work and the general properties of cathode materials, one can conclude that hopping of electrons is the fastest process of charge transport occurring in the studied samples. Consequently, electric conduction is dominated by the electronic component of total conductivity, and the ionic component is strongly suppressed. It seems that enhancement of electronic conductivity after HPHT treatment (Figure 4) has a quite natural and intuitive interpretation. It can be seen from Formula (Equation 3) that electronic conductivity depends (like $\frac{1}{R}$) on the average distance between hopping centers (Fe${}^{2+}$–Fe${}^{3+}$). Similarly, the activation energy $E_{a}$ of electron hopping decreases when *R* shortens, because $E_{a}$ = const.$\cdot (1-\frac{r_{p}}{R})$, where $r_{p}$ denotes a radius of small polaron [14]. This is consistent with the expected reduction of distances between atoms after high-pressure treatment. In disordered systems containing iron, electron hopping occurs via oxygen bridge between Fe${}^{2+}$ and Fe${}^{3+}$ centers [11]. The concentration of such aliovalent ions in nanomaterials is much higher than in glasses and polycrystalline materials, and therefore, hopping is much more effective in such systems. This is effect of a large surface-to-grain volume ratio in nanomaterials. An external high pressure additionally increases this effect because of reduction of distance between Fe${}^{2+}$/Fe${}^{3+}$ redox couples. The observed reduction in relaxation times (Figure 7) occurred because of the same mechanism. It is seen from comparison of Figure 4 and Figure 7 that the values of activation energies determined from electronic conductivity and relaxation time measurements were almost the same. An effect of high pressure was also observed in the case of Na${}^{+}$ jumps. Relaxation time and corresponding activation energy considerably changed after HPHT treatment (τ decreased by one order of magnitude, and $E_{a}$ decreased from 0.52 to 0.28 eV in the middle range of temperature; cf. Figure 7). As a result of high-pressure-high-temperature treatment, the initial glass transformed into a two-phase nanocomposite-consequently, grain boundaries appeared in the studied sample. This caused a non-Arrhenius behavior of the process related to inter-grain transport (green plot in Figure 7). Additionally, intra-grain (bulk) ionic motion exhibited two activation energies, apparently due to two nanocrystalline phases induced by HPHT treatment.

## 5. Conclusions

A new preparation method of cathode materials for sodium batteries was proposed by using high-pressure-high-temperature treatment of initial glass of composition NaFePO${}_{4}$. Applying a pressure of 1 GPa effectively enhanced the electrical conductivity of the studied material and induced two electrochemically active phases—NASICON and alluaudite. It was shown that the effect of high pressure was permanent and lasted for several months. Another advantage of high-pressure cathode materials is their anticipated higher volumetric capacity compared with standard cathodes with carbon additions. It was demonstrated that the combination of high pressures and broadband dielectric spectroscopy is a powerful method to resolve various relaxation processes and molecular movements in solids. Particularly, in this study it was possible to separate electronic and ionic components of electric charge movement. 

## Figures and Tables

**Figure 1 nanomaterials-13-00164-f001:**
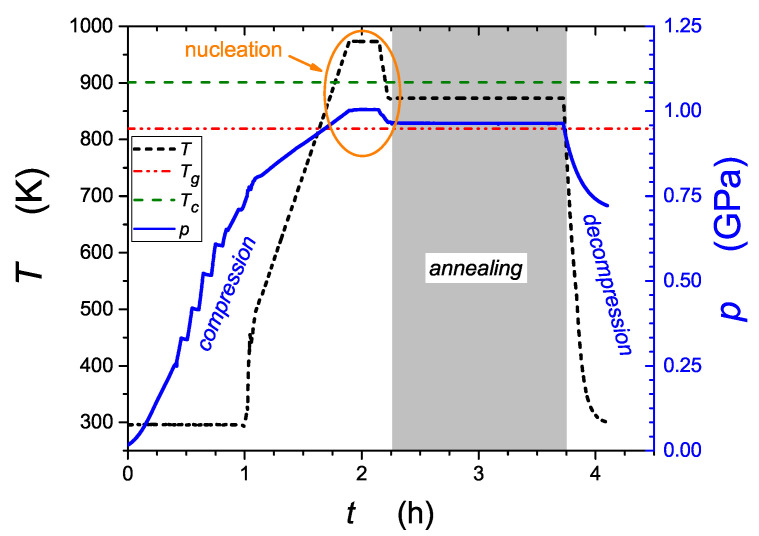
The course of high-pressure–high-temperature treatment (HPHT) used in this study. After 2 h of compression, the annealing process started. Above the crystallization temperature ($T_{c}$), the nucleation takes place.

**Figure 2 nanomaterials-13-00164-f002:**
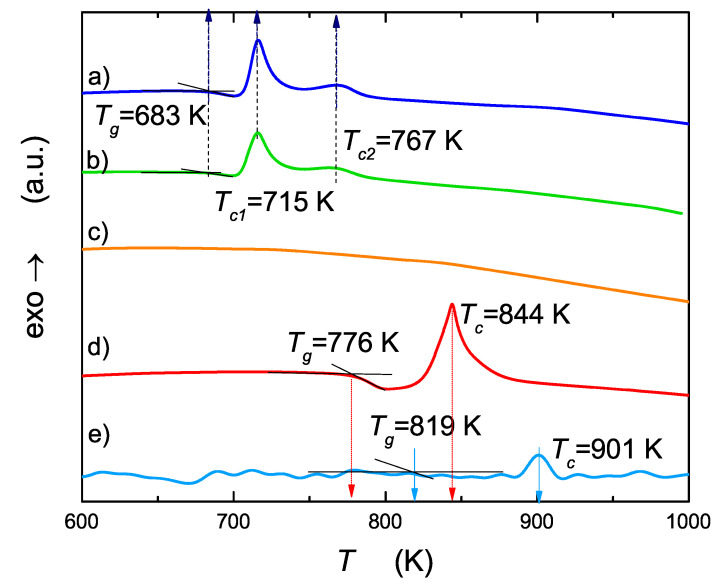
DTA runs of studied samples: (a) NaFe${}_{0.85}$V${}_{0.10}$PO${}_{4}$ glass quenched on a steel plate (NFV/st), (b) NaFe${}_{0.85}$V${}_{0.10}$PO${}_{4}$ glass quenched on a copper plate (NFV/co), (c) NaFePO${}_{4}$ quenched on steel (NF/st), (d) NaFePO${}_{4}$ glass quenched on copper (NF/co), (e) NaFePO${}_{4}$ glass quenched on copper (NF/co/HPHT), measured under pressure of 1 GPa.

**Figure 3 nanomaterials-13-00164-f003:**
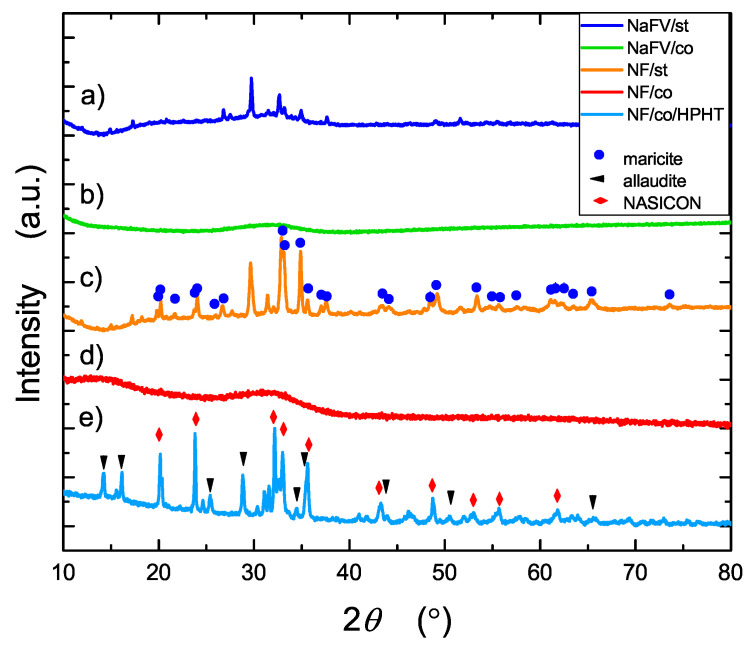
XRD patterns of the studied samples: (a) NaFe${}_{0.85}$V${}_{0.10}$PO${}_{4}$ glass quenched on steel plate (NFV/st), (b) NaFe${}_{0.85}$V${}_{0.10}$PO${}_{4}$ glass quenched on a copper plate (NFV/co), (c) NaFePO${}_{4}$ sample quenched on steel (NF/st), (d) NaFePO${}_{4}$ glass quenched on copper (NF/co), (e) NaFePO${}_{4}$ glass after high-pressure–high-temperature treatment (NF/co/HPHT). Bragg peaks in patterns (c) and (e) correspond to identified crystalline phases (see text).

**Figure 4 nanomaterials-13-00164-f004:**
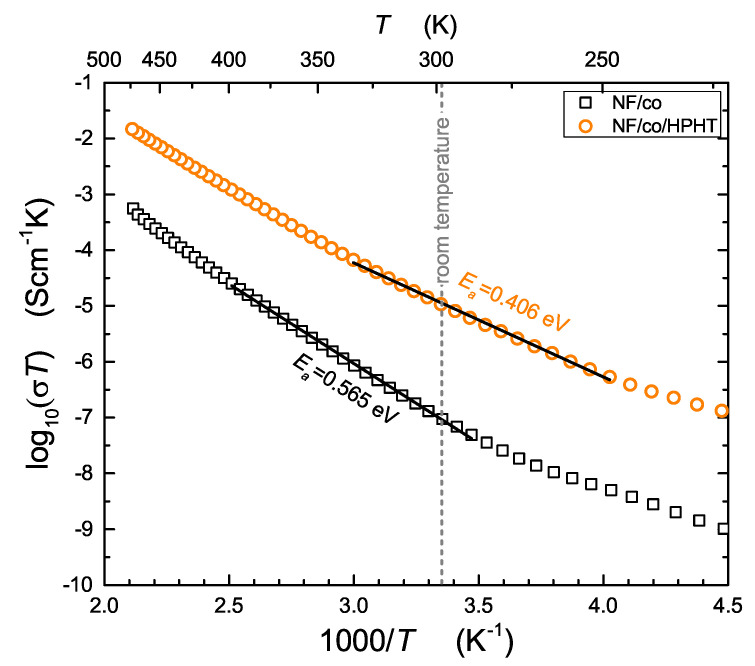
Temperature dependencies of electric conductivity for: NaFePO${}_{4}$, glass quenched (NF/co), and the corresponding nanocomposite after HPHT treatment (NF/HPHT).

**Figure 5 nanomaterials-13-00164-f005:**
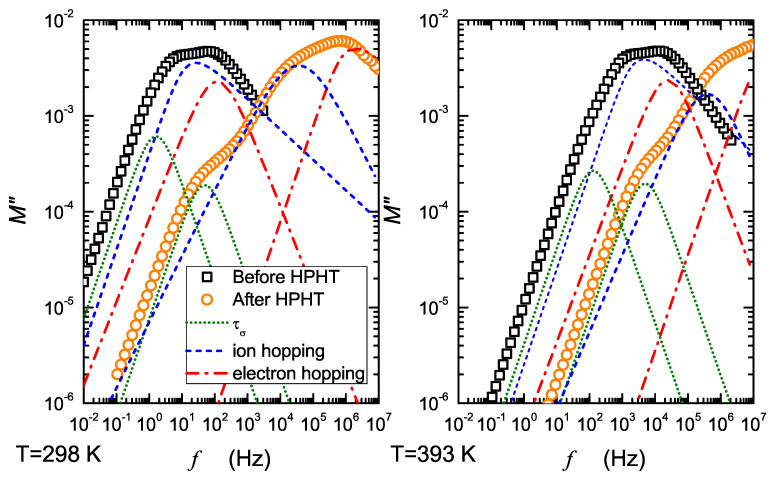
Resolved spectra of the imaginary part of the electric modulus for the studied samples before and after HPHT treatment at 298 and 393 K.

**Figure 6 nanomaterials-13-00164-f006:**
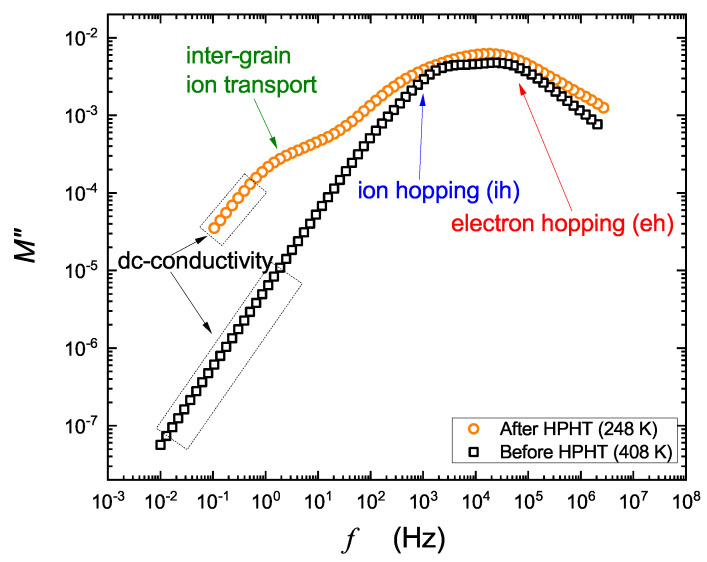
Temperature-time-scaling (TTS) for studied samples before and after HPHT treatment. Inter-grain ion transport, intra-grain ion hopping (ih) and electron hopping (eh) are resolved. Note that curves correspond to 248 and 408 K, which is a 160 K shift.

**Figure 7 nanomaterials-13-00164-f007:**
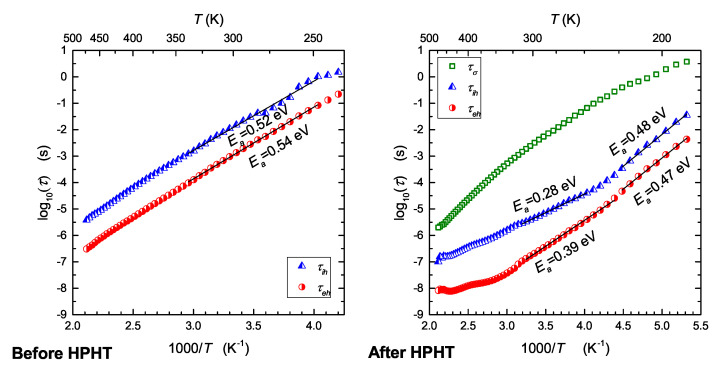
Temperature dependencies of relaxation time for various processes revealed in measurements of electric modulus, before and after HPHT treatment.

**Table 1 nanomaterials-13-00164-t001:** Glass transition ($T_{g}$) and crystallization ($T_{c1}$, $T_{c2}$) temperatures of studied samples.

Sample	$T_{g}$ [K]	$T_{c1}$ [K]	$T_{c2}$ [K]
(a) NFV/st	683	715	767
(b) NFV/co	683	715	767
(c) NF/st	-	-	-
(d) NF/co	776	844	-
(e) NF/co/HPHT	819	901	-

**Table 2 nanomaterials-13-00164-t002:** Contents of crystalline phases detected in the studied samples.

Phase/Structure	NF/st *	NF/co **	NF/co/HPHT
(Ref. Code)	(Crystallite Size [nm])	(Crystallite Size [nm])	(Crystallite Size [nm])
Na${}_{3}$Fe${}_{2}$(PO${}_{4}$)${}_{3}$/NASICON	-	-	44.7%
(ICCD 04-011-4360)			(91 nm)
Na${}_{2}$Fe${}_{3}$(PO${}_{4}$)${}_{3}$/alluaudite	-	-	55.3%
(ICCD 04-009-8653)			(42 nm)
NaFePO${}_{4}$/maricite	100%	-	-
(ICCD 04-012-9665)	(39 nm)		

* maricite crystaline phase, ** pure amorphous material.

## Data Availability

Data are available from authors on reasonable request.

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
