# Peer review of "Novel High-Pressure Nanocomposites for Cathode Materials in Sodium Batteries"

_nanomaterials, 2022, doi:10.3390/nano13010164_

Round 1

Reviewer 1 Report

This research paper reported synthesis of high-pressured nanocomposites for cathode materials in sodium batteries. This research showed that high-pressure cathode materials may exhibit higher volumetric capacities compared with commercially used cathodes with carbon additions. This paper can be accepted after addressing the following questions.

1. There are typos that need to be corrected and united.

2. How can you be sure of the composition of NaFe0.85V0.10PO4?

3. It would be better representing an image of synthesized NaFe0.85V0.10PO4 particles.

4. Since crystallinity of cathode material is important, It would be better representing NaFe0.85V0.10PO4 high-resolution TEM data.

5. It would be better representing cycle performance of synthesized NaFe0.85V0.10PO4 particles.

Author Response

We are grateful for all comments. During second reading and re-writing all reviewer’s suggestions have been considered. Please find our answers and comment below. We hope the paper will be suitable for publication in Nanomaterials in current form.

  1. There are typos that need to be corrected and united.

All typos and language mistakes have been corrected and clarified.

  1. How can you be sure of the composition of NaFe0.85V0.10PO4?

Please kindly notice, that obtained material was a glass of quaternary system: Na2O-FeO-V2O5-P2O5 . The proposed composition of aforementioned glass, was a nominal composition of substrates used at the synthesis process of glass (0.5Na2O-0.85FeO-0.05V2O5-0.5P2O5). The DTA analysis of chemical reactions occurring during synthesis showed no loss related to: Na, P or V evaporation. Our thermal studies cross-referenced with XRD on used materials (NaFe0.85V0.10PO4 and NaFePO4) proved, that it was possible to obtain fully amorphous material of nominal composition NaFePO4 (without vanadium additive). Therefore, further studies were performed for NaFePO4 glasses, for which measurements of NaFe0.85V0.10PO4 glasses remained as a reference. Some controversy related to non-normalized composition of that glass one can remove, if we take into consideration that except of homogeneous doping with V (V substitutes Fe), we also have heterogeneous doping (V in the interstitial sites).

Please find line 72 which now states: The nominal composition of such Na2O-FeO-V2O5-P2O5 heterogeneously doped glasses was NaFe0.85V0.10PO4.

  1. It would be better representing an image of synthesized NaFe0.85V0.10PO4 particles.

Thank you for this suggestion. Vanadium doped sample was just a comparative sample, this is why we decided not to add particle images. Please find additional explanation at lines (139-143):

“Singular peaks visible at curve a in figure 3 may originate from vanadium iron oxide and iron oxide crystalline nuclei.  Curve b in figure 3 shows improvement of quality in glassy material obtained from NaV0.10Fe0.85PO4 with application of copper plate. This confirmation led us to synthesis of pure glassy NaFePO4 (curve d in figure 3) which was used in further experiments.”

  1. Since crystallinity of cathode material is important, It would be better representing NaFe0.85V0.10PO4 high-resolution TEM data.

In this study vanadium doped samples were used as reference material for main measurements focused on NaFePO4. TEM studies are expensive and time consuming and in fact we expect similar images to those found by us earlier for lithium analogues. Please find examples of such high-resolution TEM data for those samples in: T. Pietrzak, M. Wasiucionek, I. Gorzkowska, J. Nowiński, J,.Garbarczyk, "Novel vanadium-doped olivine-like nanomaterials with high electronic conductivity", 2013 (https://doi.org/10.1016/j.ssi.2013.02.012) or in Ref. [12].

  1. It would be better representing cycle performance of synthesized NaFe0.85V0.10PO4 particles.

Please kindly notice that NFVP sample was shown in this paper for a reference and the main topic of this paper was NFP composition, because we managed to synthesis glassy sample of nominal NaFePO4 composition, i.e. without supporting V2O5 glass former additive. We will be very happy to investigate electrochemical properties of NFP material after HPHT treatment. However, decent electrochemical characterisation requires much time to complete cycles at various rates and with enough number of cycles. This is out of scope of this paper, which was focused on electrical phenomena related to high-pressure high-temperature nanocrystallization of glasses. We hope to provide electrochemical data in the incoming publication.

Reviewer 2 Report

The manuscript submitted by Szpakiewicz-Szatan et al. to the journal of Nanomaterials discussed a method to improve the electrical conductivity of NaFePO4 after high-temperature high-pressure treatment. The research is innovative, especially by combining with broadband dielectric spectroscopy measurement. The paper was well-written, and I would recommend publishing this paper, however, the authors need to address the following questions/comments.

1. Line 107, what is the purpose of annealing the materials at 873 K, below Tc at measured pressure? what would happen if the materials were annealed at 974 K instead before quenching?

2. Please explain the difference in XRD patterns of NaFV/st and NaFV/co, why one crystalline and one amorphous? Also, what phase do the authors expect for NaFV/st?

3. Line 148, for the HPHT treatment sample, which phase is preferable Na3Fe2(PO4)3 or Na2Fe3(PO4)3? Does the author know if the ideal phase can be made using a similar method with excess Na3PO4 or FePO4?

4. Any explanation for why the increase in conductivity for Na composites is higher than for the Li analogs?

5. It is very cool to see broadband dielectric spectroscopy measurements suggesting the increase of electronic conductivity, the authors also suggest a carbon coating-free process might be valid for these materials. But does the author ever make cells out of the materials to really test if really work in a real cell or battery?  

Author Response

We are grateful for all comments. During second reading and re-writing all reviewer’s suggestions have been considered. Please find our answers and comment below. We hope the paper will be suitable for publication in Nanomaterials in current form.

  1. Line 107, what is the purpose of annealing the materials at 873 K, below Tc at measured pressure? what would happen if the materials were annealed at 974 K instead before quenching?

Annealing temperature below Tc (and above Tg) was performed to proceed nanocrystallization. This path has been tested for many years in our group with different genders of materials. Such a decision was made based on our previous results with Li analogues – see for example Ref. [18] and preprint Starzonek, S., Rzoska, S. J., Drozd-Rzoska, A., Bockowski, M., Pietrzak, T. K., Garbarczyk, J. E. (2022). Pressure-driven relaxation processes in nanocomposite ionic glass LiFe0.75V0.10PO4 (https://doi.org/10.48550/arXiv.2206.13300). If a material was annealed for longer time at temperature above Tc further crystalline growth would occur which would result in larger crystallites.

Please find lines 106-112 which now state:

“In order to initiate nucleation of nanocrystallites from amorphous phase, the NF/co glass sample was subjected to 1 GPa pressure, then heated up to 974 K (i.e. above Tc measured under pressure) and annealed at this temperature for 15 minutes. Next the sample was cooled down to 873 K (i.e. between Tg and Tc measured under pressure) and annealed for further 1.5 hour in order to reduce entropy. Finally, it was cooled down to room temperature while decompressed to atmospheric pressure. Our long-time monitoring showed that the effect of HPHT on samples properties (conductivity) lasted for months.”

  1. Please explain the difference in XRD patterns of NaFV/st and NaFV/co, why one crystalline and one amorphous? Also, what phase do the authors expect for NaFV/st?

NaFV/st sample was amorphous with slight addition of crystalline impurities (as cross referenced with Differential Thermal Analysis), whilst NaFV/co was purely amorphous. Crystalline impurities observed as singular peaks on XRD figure may be expected from crystal nuclei of FeV2O4, Fe3O4 and Fe2O3 phases. Please find additional explanation

in text in lines 140-144 in which we added:

“Singular peaks visible at curve a in figure 3 may originate from vanadium iron oxide and iron oxide crystalline nuclei.  Curve b in figure 3 shows improvement of quality in glassy material obtained from NaV0.10Fe0.85PO4 with application of copper plate. This confirmation led us to synthesis of pure glassy NaFePO4 (curve d in figure 3) which was used in further experiments.”

  1. Line 148, for the HPHT treatment sample, which phase is preferable Na3Fe2(PO4)3 or Na2Fe3(PO4)3? Does the author know if the ideal phase can be made using a similar method with excess Na3PO4 or FePO4?

Preferable would be alluaudite (Na3Fe3(PO4)3) phase, as it is a composition containing more Fe atoms (which ions are source of conducting electrons) over Na atoms. Other research (Ref. [15], NASICON and Ref. [21], alluaudite) have shown that conductivity of alluaudite phase is higher than NASICON phase (when synthesised at atmospheric pressure).

Please find in lines 208-210 corrected conductivity of glassy NASICON and alluaudite quoted from references mentioned above:

“For comparison, room temperature conductivities of NASICON and tri-iron alluaudite glass are equal to 1.33·10-11 S/cm[15] and 2.87·10-9 S/cm[21] respectively.”

Thank you for inquisitive question. Finding whether such an ideal phase could be obtained will be one of the research goals in the coming years as part of the grant we have recently received.

  1. Any explanation for why the increase in conductivity for Na composites is higher than for the Li analogs?

Li analogues crystalised into LiFePO4 and Li3Fe2(PO4)3, we may expect alluaudite phase (Na2Fe3(PO4)3) to exhibit higher conductivity than triphylite phase. Vanadium dopant (which was not detected in crystalline phases at Li analogues) possibly remained part of glassy matrix which exhibited lower conductivity than crystalline phases.

  1. It is very cool to see broadband dielectric spectroscopy measurements suggesting the increase of electronic conductivity, the authors also suggest a carbon coating-free process might be valid for these materials. But does the author ever make cells out of the materials to really test if really work in a real cell or battery?  

We will be very happy to investigate electrochemical properties of NFP material after HPHT treatment. However, decent electrochemical characterisation requires much time to complete cycles at various rates and with enough number of cycles. This is out of scope of this paper, which was focused on electrical phenomena related to high-pressure high-temperature nanocrystallization of glasses. We hope to provide electrochemical data in the incoming publication.